# Genome-Wide Association Study and Identification of Candidate Genes Associated with Seed Number per Pod in Soybean

**DOI:** 10.3390/ijms25052536

**Published:** 2024-02-22

**Authors:** Qiong Wang, Wei Zhang, Wenjing Xu, Hongmei Zhang, Xiaoqing Liu, Xin Chen, Huatao Chen

**Affiliations:** 1Institute of Industrial Crops, Jiangsu Academy of Agricultural Sciences, Nanjing 210014, China; wqwzw@163.com (Q.W.);; 2Zhongshan Biological Breeding Laboratory (ZSBBL), Nanjing 210014, China

**Keywords:** soybean, seed number per pod, leaf characteristics, genome-wide association study, single nucleotide polymorphism

## Abstract

Soybean (*Glycine max* [L.] Merr.) is one of the primary sources of plant protein and oil for human foods, animal feed, and industrial processing. The seed number per pod generally varies from one to four and is an important component of seed number per unit area and seed yield. We used natural variation in 264 landraces and improved cultivars or lines to identify candidate genes involved in the regulation of seed number per pod in soybean. Genome-wide association tests revealed 65 loci that are associated with seed number per pod trait. Among them, 11 could be detected in multiple environments. Candidate genes were identified for seed number per pod phenotype from the most significantly associated loci, including a gene encoding protein argonaute 4, a gene encoding histone acetyltransferase of the MYST family 1, a gene encoding chromosome segregation protein SMC-1 and a gene encoding exocyst complex component EXO84A. In addition, plant hormones were found to be involved in ovule and seed development and the regulation of seed number per pod in soybean. This study facilitates the dissection of genetic networks underlying seed number per pod in soybean, which will be useful for the genetic improvement of seed yield in soybean.

## 1. Introduction

Soybean (*Glycine max* [L.] Merr.) is an important agronomical and economical crop that provides plant-based protein and oil for human consumption [1]. Cultivated soybean was domesticated from its wild progenitor (*Glycine soja* Sieb. & Zucc.) approximately 5000 years ago in China [2]. Yield is always one of the most important objectives in crop breeding and in soybean, determined by seed number per unit area and seed weight. The seed number per unit area is a product of the number of plants per unit area, the number of pods per plant, and the seed number per pod (SNPP) [3]. Soybean plants produce four main types of pods, varying from one to four seeds per pod. The seed number per pod is regarded as one of the important determinants that determine soybean yield and has long been considered in soybean production [4,5]. To date, over 290 quantitative trait loci (QTL) related to seed weight in soybean have been identified across 20 chromosomes (http://www.soybase.org (accessed on 13 December 2023)). However, there have been few reports of research on SNPP traits. *Ln* is a key regulator of leaf shape and seed number per pod in soybean [6,7]. The transition from broad (*Ln*) to narrow leaflet (*ln*) is associated with an amino acid substitution in the EAR motif of the *GmJAG1* gene encoded by *Ln*, which is homologous to *Arabidopsis JAGGED* (*JAG*) that regulates lateral organ development and the variant exerts a pleiotropic effect on fruit patterning [6]. The narrow leaf accession Lvbaoshi produces a higher number of seeds per pod (with a high ratio of 4-seed pod and 3-seed pod) than the broad leaf accession Han2296 (with a high ratio of 2-seed pod and 1-seed pod). An F2 mapping population with 1868 individual lines derived from a cross between these two lines revealed *Glyma20g25000* (homologous to *Arabidopsis JAG*) as the causal gene of the *Ln* locus [7].

Most soybean varieties produce only 2 ovules but more than 3000 pollen grains per flower, 1500 times that of ovules [8]; therefore, the number of ovules (i.e., female gametes) sets the maximum seed number per pod in soybean. Recombinant inbred populations derived from intraspecific reciprocal crosses of three cultivars (Noir 1, Minsoy and Archer) revealed that the SNPP in soybean is determined by multiple QTL that accounted for ~50% of the heritable variation [9]. In addition to a QTL on U3 linked to *Lf1* that determined leaflet number and a QTL on U17 linked to *Ln* for leaflet shape, a major QTL on Linkage Group 13 is located in a region known to contain Ms1, Ms6, and St5 loci for male and female sterility [10,11], suggesting that these loci may influence the number of ovules per pod and consequently the number of seeds per pod.

Genome-wide association study (GWAS) is a powerful approach for identifying the genetic architecture underlying variability in complex traits in crops based on phenotypes and genotypes association. GWAS make use of high genome coverage of single nucleotide polymorphisms (SNP) markers and has been very effective for identifying the loci or putative causal genes controlling the complex traits in crops. In this study, we focused on the genetics underlying the SNPP trait variation in soybean, a total of 264 representative soybean germplasms were collected, and genome-wide association analysis for the SNPP trait was conducted. We integrated the GWAS approach with expression profiling data, functional annotation of the orthologs of *Arabidopsis*, and annotation of SNPs to identify candidate loci and genes. This study provides new insights into the genetic control of soybean SNPP, laying the foundation for molecular breeding to develop soybean cultivars with increasing yield.

## 2. Results

### 2.1. Phenotypic Variations of the SNPP Trait

We phenotyped a GWAS panel composed of 264 diverse accessions of soybean planted in Nanjing City, Jiangsu Province for two consecutive years (2022 and 2023, NJ22 and NJ23 for short). In order to clarify the phenotypic variation of the SNPP trait, we analyzed pod type in 264 accessions under two environments. The average number of seeds per pod of all accessions varied from 1.56 to 3.40 (average 2.42) in 2022 and ranged from 1.62 to 3.35 (average 2.48) in 2023 (Figure 1). The results showed that the SNPP trait was normally distributed, indicating that it is a quantitative trait, controlled by multiple genes. An average SNPP of 2.2–2.5 was the most frequent in the two environments. In addition, a significant positive correlation was observed between the SNPP in NJ22 and NJ23, with a correlation coefficient *r* = 0.43.

The number of seeds per pod in soybean is tightly associated with the leaf shape trait [4]. Leaf shape in soybean can be mainly classified into two categories: ovate and narrow leaflet. Different leaf morphologies may exist in the upper and lower part of a single soybean plant, which is known as heterophylly. Current reports are mainly focused on studying single types of leaf morphology and very few studies have considered the heterophylly in soybean. We employed correlation analysis to evaluate phenotypic correlations among the average SNPP trait and leaf characteristics traits including upper leaf length, upper leaf width, upper leaf shape, lower leaf length, lower leaf width, lower leaf shape, and heterophylly leaf index that described in our previous research, where leaf shape of both the upper and lower leaves were obtained by calculating the ratio of leaf length to leaf width and heterophylly leaf index was calculated by dividing upper leaf shape by lower leaf shape [12]. As a result, the SNPP trait had significant positive correlations with upper leaf shape and lower leaf shape (Figure 2). These results are consistent with previous reports that narrow leaflet soybean varieties usually have more seeds per pod than ovate leaflet varieties [4,5,6,7,13,14,15,16]. At the same time, the SNPP trait showed a significant positive correlation with the heterophylly leaf index. These data indicated that the SNPP trait may be influenced by numerous genes, genes that are not associated with leaf characteristics traits may also be involved in the regulation of the number of seeds in soybean.

### 2.2. Genome-Wide Association Analysis

Phenotypic data and NGS data were combined to investigate the genetic factors controlling the number of seeds per pod in soybean. The overall performances of the 264 accessions were predicted as the best linear unbiased prediction (BLUP) implemented in the lme4 R package. Using the genotypic data of 4,750,371 high-quality SNPs with minor allele frequency (MAF) > 5% and missing rate < 10%, we identified 65 associated loci (Figure 3; Appendix A). Among them, 11 could be detected in multiple environments (Table 1). These signals were located on chromosomes 3, 4, 5, 6, 7, 10, and 13, respectively, indicating the presence of candidate genes associated with the average SNPP at these loci. The locus detected on chromosome 5 was Chr05:17445053, and the locus located on chromosome 6 was Chr06:50457715, while the most significantly associated locus Chr13:23620712 was located on chromosome 13. Taken together, these results suggest that a number of genes influence the reproduction of soybean during the process of floral development. Loci that are detected multiple times in different environments provide a solid foundation for dissecting the gene networks underlying the SNPP in soybean.

### 2.3. Identification of Candidate Genes

The candidate regions were estimated by pairwise linkage disequilibrium (LD) correlation analysis conducted by the LDBlockShow [17]. There were 250 genes in the candidate regions, including 211 nonsynonymous and nine stop gain or stop loss nonsense mutated genes (Appendix A). We integrated the GWAS analysis with expression profiling data, functional annotation of the orthologs of *Arabidopsis*, and annotation of SNPs to identify candidate genes associated with the SNPP trait. Four candidate genes were identified to be associated with the average SNPP trait.

Locus Chr06:50428644 was identified to be associated with the average SNPP on chromosome Chr06. 27 genes were identified in the candidate region, among them, 24 contained nonsynonymous SNPs. 10 of these genes (*Glyma.06G314500*, *Glyma.06G314800*, *Glyma.06G315100*, *Glyma.06G315500*, *Glyma.06G315600*, *Glyma.06G315700*, *Glyma.06G316200*, *Glyma.06G316300*, *Glyma.06G316500*, and *Glyma.06G316600*) were significantly associated with the average SNPP, including a gene encoding protein argonaute 4 (*Glyma.06G314500*) (Figure 4a).

ARGONAUTE 4 (AGO4) is a core component of the small RNA-induced silencing pathway and may possess a meiosis-associated cellular function. AGO4 can direct DNA de novo methylation in the RNA-directed DNA Methylation (RdDM) pathway together with its orthologs AGO6 and AGO9 [18,19,20]. In *Arabidopsis*, AGO4 influences meiosis. Meiotic defects have been observed in the RdDM pathway mutants ago4 [21]. AGO9 plays a central role in controlling cell fate in the *Arabidopsis* ovule. The specification of gametophyte precursors is restricted in a non-cell-autonomous manner, resulting in the formation of female gamete [22]. The haplotype analysis identified two haplotype groups based on the nonsynonymous SNP Chr06:50342354 which was located in the gene *Glyma.06G314500*. The nonsynonymous SNP at this locus resulted in an amino acid change from threonine to alanine, and the T allele was significantly positively correlated with the SNPP (*p* = 0.02728, two-tailed *t*-test) and increased the average SNPP by 7.78% compared with the corresponding CC haplotype (Figure 4c). Therefore *Glyma.06G314500* was identified as a candidate gene for the locus Chr06:50428644. Using the expression profiling data obtained from the publicly available database (https://phytozome-next.jgi.doe.gov (accessed on 19 October 2023)), low expression of this gene was observed (Figure 4b), indicating that the gene may not be regulated at the transcript level.

In addition, based on the functional annotations and expression pattern data of these genes for the locus Chr06:50428644, we identified another candidate gene *Glyma.06G316400*. The expression of Glyma.06G316400 was highest in the flower tissues (Figure 4b). *Glyma.18G012600* was annotated as a homolog of the histone acetyltransferase of the MYST family 1 encoding gene *HAM1*. *HAM1* is essential for male and female gametophyte development in *Arabidopsis*, expressed mainly in growing organs such as shoots and flower buds [23]. So, *Glyma.18G012600* was regarded as the other candidate gene responsible for the average SNPP locus Chr06:50428644.

We also identified a major locus that is associated with the average SNPP on chromosome Chr13. Based on the expression pattern data and the annotations of genes, two candidate genes were identified at adjacent positions within the association signal region (Figure 5a), including a gene encoding chromosome segregation protein SMC-1 (*Glyma.13G123700*) and a gene encoding exocyst complex component EXO84A (*Glyma.13G125800*). Both genes are predominantly highly expressed in flowers (Figure 5b). *Glyma.13G123700* is annotated as encoding a structural maintenance of chromosomes protein 1 that is homologous to *Arabidopsis SMC1*. *SMC1* is required for chromosome segregation and cell division during embryogenesis [24]. The SNP Chr13:23697656 located in the gene *Glyma.13G123700* resulted in an amino acid change from leucine to isoleucine. The haplotype analysis showed that the GG haplotype significantly increased the average SNPP by 15.7% compared with the TT haplotype (Figure 5c). The TT haplotype group contains 105 accessions, and GG was present in 116 accessions; while GT was very rare and appeared in only 2 accessions, it was excluded from the association analysis.

Another gene, *Glyma.13G125800* is annotated as encoding an exocyst complex component EXO84A that is homologous to *Arabidopsis EXO84A*. Floral life span determines the effective pollination period and ultimately the number of seeds. In *Arabidopsis*, *EXO84A* regulates exocytotic compartment degradation and stigma senescence, the *exo84c* mutant showed a prolonged effective pollination period and higher seed sets [25]. A nonsynonymous SNP (T/G) in *Glyma.13G125800* resulted in an amino acid change from aspartic acid to glutamic acid. Two major haplotypes based on the nonsynonymous SNP Chr13:23915184 were associated with the GWAS signal for the average SNPP. The TT haplotype group contains 91 accessions, and GG was present in 128 accessions; while TG was very rare and appeared in only 3 accessions, it was excluded from the association analysis. Individuals carrying TT produced more seeds per pod than the GG haplotype-carrying accessions by 9.3% (Figure 5c).

Moreover, four genes were identified to be related to auxin biosynthesis or participate in the auxin signaling pathway in the candidate gene set of association signal on chromosome 13, including a gene encoding protein WEAK AUXIN RESPONSE 1 (*Glyma.13G123400*), a gene encoding patellin-4 (*Glyma.13G123600*), a gene encoding auxin-responsive protein IAA11 (*Glyma.13G127000*) and a gene encoding protein ENHANCER OF TIR1-1 AUXIN RESISTANCE 3 (*Glyma.13G127800*). Among them, *Glyma.13G123600* is annotated as encoding a patellin-4 that is homologous to *Arabidopsis PATL4*. Previous studies on *Arabidopsis* demonstrated that *PATL4* was involved in the cellular response to auxin stimulus and protein localization involved in auxin polar transport [26]. By analyzing the expression pattern of *Glyma.13G123600*, we found that this gene was predominantly highly expressed in flower tissues (Figure 5b). These results indicate that *Glyma.13G123600* may be involved in the regulation of the number of seeds in soybean by performing functions in the auxin pathway. Therefore, we identified these three genes as candidate genes for the significant signal on chromosome 13.

## 3. Discussion

The number of seeds per pod is tightly associated with the leaflet-shaped trait [4]. The pleiotropic *Ln* locus (encoded GmJAG1) regulated leaf shape and pod type, an amino acid substitution in the ERF motif of GmJAG1 increased the number of seeds per pod, and increased soybean seed yield in turn [6,7]. However, the molecular basis of the relationship between leaf/flower development and pods or seeds, which are derived from the ovules, is poorly understood. The SNPP trait may be influenced by numerous genes, genes that are not associated with leaf characteristics traits may also participate in the controlling of the number of seeds in soybean. Using 264 landraces and improved cultivars or lines of soybean, we investigated the genetic basis of the complex SNPP trait in soybean. We employed GWAS, integrating phenotype information and SNP genotype data, and proposed numerous key loci or genes for further investigation and verification.

Plant hormones are involved in gynoecium development and can affect ovule number. Ovule number and gynoecium size are two fundamental processes that affect seed yield [27]. In *Arabidopsis*, ovules are initiated from the carpel margin meristem (CMM) during floral development [28]. The phytohormone auxin has wide-ranging effects on growth and development. Auxin biosynthesis, signaling, and transport mutants have pleiotropic effects on gynoecium development, and affect tissues and organs derived from the CMM [29]. In our study, six genes were identified to be related to auxin biosynthesis or the response to auxin stimulus in the candidate gene set (Appendix A), including a gene encoding SAUR-like auxin-responsive protein family (*Glyma.03G029600*), a gene encoding cytochrome P450 83B1 (*Glyma.03G030800*) that functions in auxin homeostasis, a gene encoding protein WEAK AUXIN RESPONSE 1 (*Glyma.13G123400*), a gene encoding patellin-4 (*Glyma.13G123600*), a gene encoding auxin-responsive protein IAA11 (*Glyma.13G127000*) and a gene encoding protein ENHANCER OF TIR1-1 AUXIN RESISTANCE 3 (*Glyma.13G127800*). Among them, *Glyma.13G123600* is annotated as encoding a Patellin-4 that is homologous to *Arabidopsis PATL4*, which was confirmed to be involved in the cellular response to auxin stimulus and protein localization involved in auxin polar transport [26]. By analyzing the expression profile data of *Glyma.13G123600*, we found that this gene exhibited predominantly high expression in flower tissues (Figure 5b). This result indicates that *Glyma.13G123600* may regulate the SNPP by performing functions in the auxin pathway.

Furthermore, gibberellins (Gas) were shown to negatively regulate ovule number in *Arabidopsis* [30]. The *Arabidopsis* SAUR36 integrates auxin and gibberellin signals to regulate hypocotyl elongation [31]. In this study, *Glyma.03G029600* was identified as a homolog of the SAUR-like auxin-responsive protein family encoding gene SAUR59, and may also be involved in the gibberellin signaling pathway. Brassinosteroids (BR) play important roles in the growth and development of plants, and may also participate in ovule initiation and development in *Arabidopsis* [32]. In the present study, two genes related to BR signaling were identified, including a gene encoding serine/threonine-protein kinase BSK3 (*Glyma.03G011000*) that acts as a positive regulator of BR signaling [33] and a gene encoding brassinosteroid-responsive RING protein 1 (*Glyma.07G068000*). These findings suggest that the seed number per pod trait may be regulated by different loci that spread out across chromosomes. Cytokinin has also been implicated in ovule initiation and development in *Arabidopsis*, double mutants in the cytokinin degrading cytokinin oxidases/dehydrogenases (CKXs) produced more than double the number of ovules as in wild-type controls [34]. In the current study, genes participating in the cytokinin signaling pathway that are associated with the SNPP trait could not be identified in our GWAS around the peak loci signals.

The cytochrome P450 family is one of the biggest protein families in plants, and the cytochrome P450 genes constitute up to 1% of the total gene annotations in several sequenced angiosperms [35] (Nelso, 2004). Several cytochrome P450 genes have been reported to be involved in the biosynthesis and catabolism of phytohormones [36] (Mizutani, 2010). CYP701A and CYP88A regulate GA biosynthesis to influence seed germination and shoot growth in Arabidopsis [37,38] (Helliwell, 1998, 2001). In rice, the CYP724B1 could control grain size by affecting BR-related genes [39,40,41] (Shi, 2015, Zhou Zhou 2017). The cytochrome P450 genes also play a key role in regulating seed size in soybean. It was reported that the CYP78A10 may play an important role in the seed development of soybean [42] (Wang. 2015). The soybean CYP78A10 gene underwent artificial selection during domestication and improvement. Two highly conserved alleles (CYP78A10a and CYP78A10b) were identified in wild soybeans, landraces, and cultivars. The CYP78A10a allele was mainly distributed in wild soybeans (Glycine soja), but was lost in the cultivars during the early stage of soybean domestication, while the CYP78A10b allele increased in frequency and became the predominant allele in cultivated soybeans. The pod number of the varieties with the CYP78A10a allele significantly increased compared with those with the CYP78A10b allele, while soybean varieties with the CYP78A10b allele were significantly heavier/bigger than those with the CYP78A10a allele. CYP78A72, which is highly conserved within terrestrial plants, also regulates seed size in soybean [43] (Zhao, 2016).

While the cytochrome P450 family plays a conserved and key role in regulating seed size in soybean, its role in regulating seed number remains largely unknown. In our study, eight cytochrome P450 genes were identified in the candidate gene set, located on chromosomes Chr03 and Chr07, respectively (Appendix A). All of these eight genes contained nonsynonymous SNPs that could induce amino acid changes. Notably, seven of the eight genes on chromosome Chr03 were located at adjacent positions, indicating the presence of important candidate genes related to soybean seed development. Among them, Glyma.03G029900 was predominantly highly expressed in flower and pod tissues, Glyma.03G030300 showed high expression in seed tissues and Glyma.03G030600 exhibited predominantly high expression in flower tissues. These genes may play key roles in regulating seed number in soybean and deserve further investigation.

The SNPP trait had significant associations with multiple genes, underscoring its genetic complexity. Our work provides a basis for the breeding of high-yield soybean cultivars and facilitates revealing the regulatory mechanisms of the SNPP trait in soybean. The GWAS results propose numerous loci and genes for further investigation. These candidate genes identified in this study should be further investigated and verified by inhibiting or overexpressing and gene-editing technologies to elucidate their detailed functions.

## 4. Materials and Methods

### 4.1. Plant Materials and Sampling

A total of 264 soybean accessions including 212 modern improved cultivars or lines and 52 landraces were collected in our study (Appendix A), representing the rich genetic diversity of soybean. All germplasms were planted for two consecutive years (2022 and 2023) in Nanjing, Jiangsu Province (NJ22 and NJ23 for short). The field planting was performed following a randomized complete block design with a single-row plot and three replications. Average SNPP was calculated as the total number of seeds divided by the number of pods.

### 4.2. Library Construction and SNP Identification

Resequencing data for each of the 264 accessions were obtained from our previously published data [44]. We aligned all paired-end sequence reads against the Williams 82 soybean reference genome Wm82.a2.v1 (https://phytozome-next.jgi.doe.gov/info/Gmax_Wm82_a2_v1 (accessed on 19 October 2023)) with BWA software (version 0.7.5a-r405) with default parameters [45]. SNP calling and validation were performed as previously described [43]. Briefly, we filtered the nonunique and unmapped reads of the generated BAM format file of the mapping results with SAMtools (version:0.1.19) [46]. The Picard program (http://broadinstitute.github.io/picard/ (accessed on 16 March 2023), version: 1.87) was applied to filter the duplicated reads for each accession. SNP calling was performed using the Genome Analysis Toolkit (GATK) version 3.8-0-ge9d806836 [47] and SAMtools software. Reads around indels from the BWA alignment were realigned with the IndelRealigner option in GATK. Only the SNPs detected by both methods were analyzed further. SNPs with a minor allele frequency (MAF) of less than 5% were discarded. As a result, a total of 4,750,371 high-quality SNPs with minor allele frequency > 5% and missing rate < 10% were obtained and used in the GWAS analysis.

### 4.3. Statistical Analysis

The distribution frequency, correlation and haplotype analysis, and variance analysis were performed using the R software package (version 3.5.1) (http://www.R-project.org (accessed on 14 November 2021)). Correlation coefficients among the SNPP trait and leaf characteristics traits were calculated using the corrplot R package (version 0.92) (https://github.com/taiyun/corrplot (accessed on 21 April 2023)). Local Manhattan plot and linkage disequilibrium heatmap for SNPs surrounding the peak loci were plotted using the R package LDBlockShow version 1.40 [17]. The expression pattern was supported by publicly available data from phytozome-next.jgi.doe.gov. For visualization of results, a heat map for candidate genes located in the candidate region was created with the pheatmap R package (version 1.0.12) (https://CRAN.R-project.org/package=pheatmap (accessed on 21 April 2023)).

### 4.4. GWAS

We carried out GWAS analysis based on the phenotypic data of the average SNPP and the genotypic data from SNPs with MAF > 0.05 and missing rate < 0.1 in the population identified in 264 soybean accessions. The association analysis was conducted using the Efficient Mixed-Model Association eXpedited (EMMAX) software package (beta version) [48] with a threshold *p*-value of 10^−5^. The kinship matrix derived from all SNPs was also calculated by EMMAX. The significance threshold was determined by the Genetic Type I error calculator (GEC) [49] and the significance threshold was estimated at approximately *p* = 10^−6^ (1.18 × 10^−6^). GWAS analysis identified 235 loci that passed the threshold and the candidate genes within these association signal regions were identified as significantly associated with the average SNPP trait.

### 4.5. Identification of Candidate Genes

Identification of candidate genes was carried out using the following strategy: we estimated the candidate region by pairwise linkage disequilibrium (LD) correlation calculated by the R package LDBlockShow [17]. Then, annotation of SNPs located in the candidate region was conducted using ANNOVAR (version: 2020-06-07) [50], based on the Wm82.a2.v1 gene set. Those nonsynonymous SNPs that can lead to amino acid changes were significantly associated with the average SNPP trait in the GWAS results, so we then examined whether the candidate genes with nonsynonymous SNPs had a dominant and/or special level of expression in the target tissue, using the expression data obtained from the publicly available database (https://phytozome-next.jgi.doe.gov (accessed on 19 October 2023)).

## Figures and Tables

**Figure 1 ijms-25-02536-f001:**
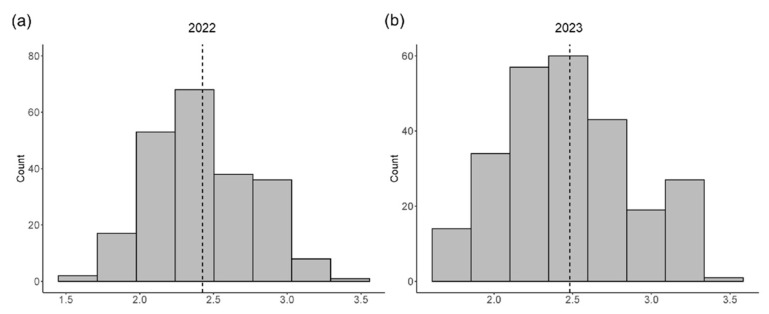
Frequency distribution of the SNPP trait in soybean: (**a**) Distribution of the average SNPP in 2022 in all accessions; (**b**) Distribution of the average SNPP in all accessions in 2023.

**Figure 2 ijms-25-02536-f002:**
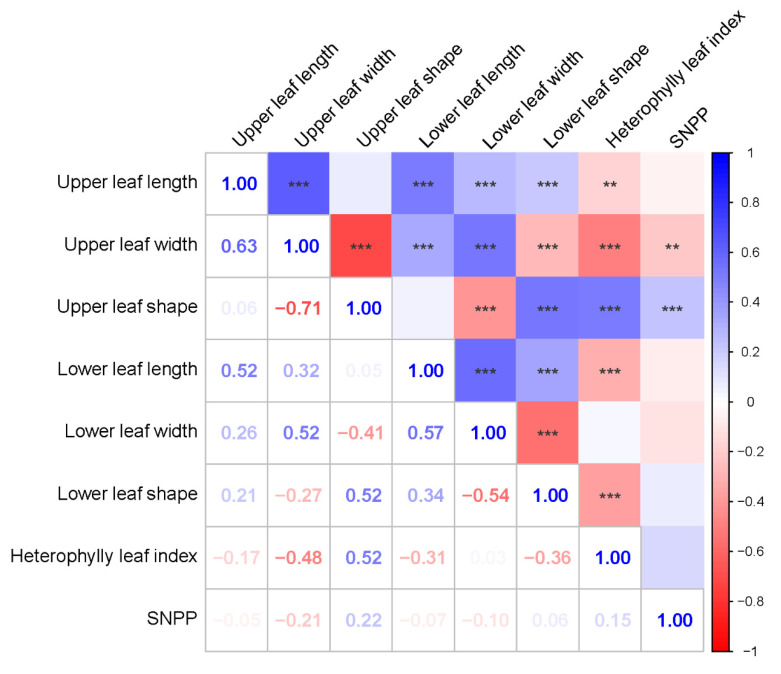
Correlation coefficients among the SNPP trait and leaf characteristics traits. ** *p* ≤ 0.01, *** *p* ≤ 0.001.

**Figure 3 ijms-25-02536-f003:**
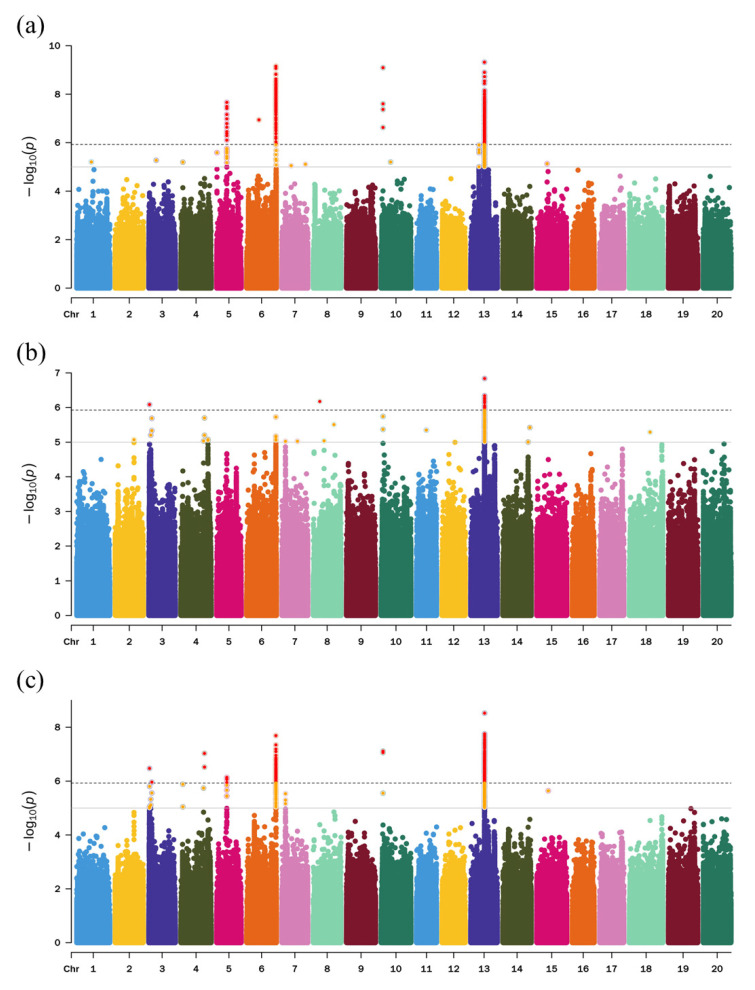
GWAS identified candidate loci associated with seed number per pod: (**a**) Manhattan plot for the average SNPP in 2022; (**b**) Manhattan plot for the average SNPP in 2023; (**c**) Manhattan plot for the average SNPP of BLUP. Chromosomes are depicted in different colors. The horizontal solid line corresponds to a −log10 (*p* values) ≥ 5 threshold and SNPs above this line are plotted as orange dots, the dashed line corresponds to −log10 (*p* values) ≥ 5.93 determined by the Genetic Type I error calculator (GEC) and SNPs above this line are plotted as red dots.

**Figure 4 ijms-25-02536-f004:**
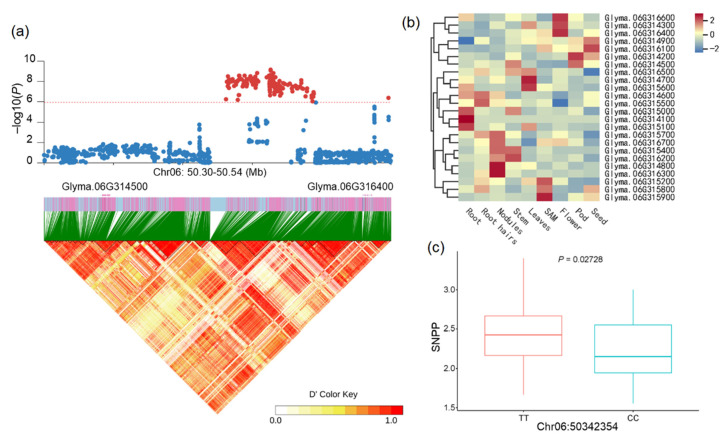
Identification of candidate genes for the associated loci Chr06:50428644: (**a**) local Manhattan plot and linkage disequilibrium heatmap of Chr06:50428644. The horizontal dashed line corresponds to a significant threshold and SNPs above this line are plotted as red dots; (**b**) expression pattern of candidate genes for Chr06:50428644; (**c**) box plots of loci Chr06:50342354. Center line, median; box limits, upper and lower quartiles; whiskers, 1.5× the interquartile range; dots, outliers. Two-sided Wilcoxon test.

**Figure 5 ijms-25-02536-f005:**
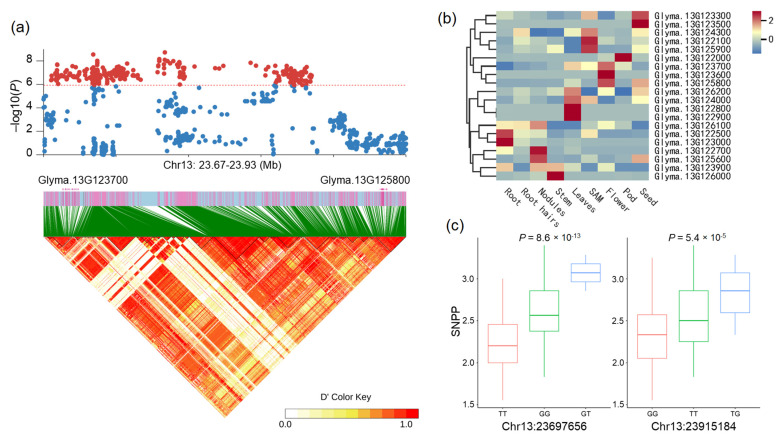
Identification of candidate genes for the significantly associated signal on chromosome 13: (**a**) local Manhattan plot and linkage disequilibrium heatmap of the most significantly associated locus. The horizontal dashed line corresponds to a significant threshold and SNPs above this line are plotted as red dots; (**b**) expression pattern of candidate genes; (**c**) box plots of loci Chr13:23697656 and Chr13:23915184. Center line, median; box limits, upper and lower quartiles; whiskers, 1.5× the interquartile range; dots, outliers. Two-sided Wilcoxon test.

**Table 1 ijms-25-02536-t001:** Seed number per pod trait-associated loci detected in multiple environments.

Signal	Peak Loci	Chromosome	Position	*p*	Environment
3_1.1	Chr03:1129563	3	1129563	3.37 × 10^−7^	NJ23, BLUP
3_3.3	Chr03:3343094	3	3343094	4.74 × 10^−6^	NJ23, BLUP
3_5.1	Chr03:5123032	3	5123032	1.10 × 10^−6^	NJ23, BLUP
4_3.4	Chr04:3411371	4	3411371	1.31 × 10^−6^	NJ22, BLUP
4_39.2	Chr04:39179195	4	39179195	1.82 × 10^−6^	NJ23, BLUP
4_40.7	Chr04:40744574	4	40744574	9.34 × 10^−8^	NJ23, BLUP
5_17.4	Chr05:17445053	5	17445053	2.18 × 10^−8^	NJ22, BLUP
6_50.5	Chr06:50457715	6	50457715	7.03 × 10^−10^	NJ22, NJ23, BLUP
7_6.2	Chr07:6182076	7	6182076	9.50 × 10^−6^	NJ23, BLUP
10_3.3	Chr10:3346012	10	3346012	8.03 × 10^−10^	NJ22, NJ23, BLUP
13_23.6	Chr13:23620712	13	23620712	4.79 × 10^−10^	NJ22, NJ23, BLUP

## Data Availability

All data presented in this manuscript are included in the Appendix A.

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
