# Peer review of "Genome-Wide Association Study and Identification of Candidate Genes Associated with Seed Number per Pod in Soybean"

_ijms, 2024, doi:10.3390/ijms25052536_

Round 1
Reviewer 1 Report
Comments and Suggestions for Authors
A manuscript entitled ” Genome-wide Association Study and Identification of Candidate Genes Associated with Seed Number Per Pod in Soybean” was reviewed for IJMS.
Here, the authors performed a GWAS study using 264 landraces against SNPP phenotype, narrowing the responsive loci to 11 regions, and more precisely, a combination of LD analysis, expression analysis, gene annotation analysis, and evaluation of SNP effects. Accordingly, they could identify some candidate genes for SNPP including meiosis-related genes, gametophyte development genes, and several plant hormone-related genes. The methodology they chose was very authentic.
However, I would like to have several questions raised.
Overall questions:
- I don’t know about the lineage relationship among the 264 landraces, but did you consider population structure before calculating GWAS within the 264 samples?
---And is it related to the result you couldn’t detect GmJAG in chromosome 20 in this GWAS?
2. You described the soybean heterophylly, indicating “SNPP had significant positive correlations with upper leaf shape, lower leaf shape, and heterophylly leaf index”. Also, SNPP was calculated as the total number of seeds divided by the number of pods (in a plant?). Does that mean your genes have some effects on SNPP for both narrow and ovate leaves? Describe somewhere in the text.
Specific question:
P1, in the first section,” The narrow leaf accession Lvbaoshi produces a higher number of seeds per pod (with a high ratio of 2-seed pod and 1-seed pod) than the broad leaf accession Han2296 (with a high ratio of 4-seed pod and 3-seed pod).” I suppose the content is contradictory to the description.
P7(Results), in Fig.7 (c), in the boxplots of loci Chr13:23697656 and Chr13:23915184, we see that hybrid haplotypes show more SNPPs than both homogeneous haplotypes. How was it explained? Heterosis?
P7(Discussion), presumed the energy a plant obtains from the sun is constant, a trade-off between some phenotypes like pod number and pod size may happen. I am curious about how the average pod size varies against the pod number distribution in your study.
Author Response
Response to Reviewer 1 Comments
Point 1: I don’t know about the lineage relationship among the 264 landraces, but did you consider population structure before calculating GWAS within the 264 samples?
---And is it related to the result you couldn’t detect GmJAG in chromosome 20 in this GWAS?
Response 1: We performed association analyses for the number of seeds per pod in 264 accessions by MLM implemented in efficient mixed-model association expedited (EMMAX) software, and kinship was derived from all SNPs. We didn't detect GmJAG1 may be because it has a pleiotropic effect on both leaf shape and pod type, and the number of seeds per pod in soybean may be influenced by numerous loci. We have provided all the germplasm used in the study in Table S3.
Point 2: You described the soybean heterophylly, indicating “SNPP had significant positive correlations with upper leaf shape, lower leaf shape, and heterophylly leaf index”. Also, SNPP was calculated as the total number of seeds divided by the number of pods (in a plant?). Does that mean your genes have some effects on SNPP for both narrow and ovate leaves? Describe somewhere in the text.
Response 2: Thanks for your advice. The SNPP trait may be influenced by numerous genes, genes that are not associated with leaf characteristics traits may also be involved in the regulation of number of seeds in soybean. We have added the content in the revised manuscript.
Point 3: P1, in the first section,” The narrow leaf accession Lvbaoshi produces a higher number of seeds per pod (with a high ratio of 2-seed pod and 1-seed pod) than the broad leaf accession Han2296 (with a high ratio of 4-seed pod and 3-seed pod).” I suppose the content is contradictory to the description.
Response 3: Thanks for your advice. We have corrected the mistakes that you pointed out in the manuscript.
Point 4: P7(Results), in Fig.7 (c), in the boxplots of loci Chr13:23697656 and Chr13:23915184, we see that hybrid haplotypes show more SNPPs than both homogeneous haplotypes. How was it explained? Heterosis?
Response 4: As the hybrid haplotypes appeared in only two and three accessions, it should be excluded from the association analysis. We have added the content in the revised manuscript.
Point 5: P7(Discussion), presumed the energy a plant obtains from the sun is constant, a trade-off between some phenotypes like pod number and pod size may happen. I am curious about how the average pod size varies against the pod number distribution in your study.
Response 5: Thanks for your advice. We will conduct correlation and association analysis for pod type, pod size and pod number in the future.
Reviewer 2 Report
Comments and Suggestions for Authors
Manuscript titled “Genome-wide Association Study and Identification of Candi-date Genes Associated with Seed Number Per Pod in Soybean” performed GWAS study on 264 soybean genotypes including 212 cultivars and 52 landraces and identified 65 associated loci among among which 11 were detected based on two years 2022 and 2023 data. The mentioned loci have been identified for seed number per pod while correlations of various leaf traits from the upper and lower canopy of the plant have been reported. The following comments below and attached in the manuscript will help to increase the quality and communication of the results the authors are intending to share through the manuscript.
1. Was there any structure among the 212 advanced materials? Check and include the information.
2. Explain the statistical analysis section in more specific detail.
3. Authors mentioned multiple environments, however only years 2022 and 2023 have been mentioned in the methods. Explain what environments are being mentioned.
4. Seed number per pod is not only the factor of leaf traits, why did authors not include other plant traits beside related to leaves?
5. Functional annotation of the candidate genes has not been performed as part of results. Include this section in results which will support most of the discussion presented in the study.
6. Please see the attached file for more comments.

Author Response
Response to Reviewer 2 Comments
Point 1: Was there any structure among the 212 advanced materials? Check and include the information.
Response 1: We performed association analyses for the number of seeds per pod in 264 accessions by MLM implemented in efficient mixed-model association expedited (EMMAX) software, and kinship was derived from all SNPs. We have provided all the germplasms used in the study in Table S3.
Point 2: Explain the statistical analysis section in more specific detail.
Response 2: Thanks for your advice. We have added the content in the revised manuscript.
Point 3: Authors mentioned multiple environments, however only years 2022 and 2023 have been mentioned in the methods. Explain what environments are being mentioned.
Response 3: The population was grown in two years, in 2022 and 2023 in Nanjing, and the overall performances of all the accessions were predicted as the best linear unbiased prediction (BLUP).
Point 4: Seed number per pod is not only the factor of leaf traits, why did authors not include other plant traits beside related to leaves?
Response 4: Thanks for your advice. The SNPP trait may be influenced by numerous genes, may also include genes that are not associated with leaf characteristics traits. The leaflet‐shape has long been suggested to be linked to the number of seeds per pod trait, however, current reports are all focused on studying single types of leaf morphology and none has considered the heterophylly. In the study reported here, we mainly investigated the correlation coefficients among the SNPP trait and heterophylly in soybean.
Point 5: Functional annotation of the candidate genes has not been performed as part of results. Include this section in results which will support most of the discussion presented in the study.
Response 5: Thanks for your advice. We have added functional annotation in results section in the revised manuscript.
Point 6: Please see the attached file for more comments.
Response 6: Thanks for your advice. We have corrected the mistakes that you pointed out in the manuscript.